# Community acceptance and social impacts of carbon capture, utilization and storage projects: A systematic meta-narrative literature review

**Jacob A. E. Nielsen** *, **Kostas Stavrianakis**, **Zoe Morrison**

Aberdeen Business School, Robert Gordon University, Aberdeen, Scotland, United Kingdom

* j.nielsen2@rgu.ac.uk

**Data Availability Statement:** All relevant data are within the article and its Supporting Information files.

## Abstract

This manuscript presents a systematic meta-narrative review of peer-reviewed publications considering community acceptance and social impacts of site-specific Carbon Capture Utilization and Storage (CCUS) projects to inform the design and implementation of CCUS projects who seek to engage with communities during this process, as well as similar climate mitigation and adaptation initiatives. A meta-narrative approach to systematic review was utilized to understand literature from a range of site specific CCUS studies. 53 peer-reviewed papers were assessed reporting empirical evidence from studies on community impacts and social acceptance of CCUS projects published between 2009 and 2021. Three separate areas of contestation were identified. The first contestation was on acceptance, including how acceptance was conceptualized, how the different CCUS projects engaged with communities, and the role of acceptance in social learning. The second contestation related to communities: how communities were represented, where the communities were located in relation to the CCUS projects, and how the communities were defined. The third contestation was around CCUS impacts and the factors influencing individuals' perceptions of impacts, the role of uncertainty, and how impacts were challenged by local communities, politicians and scientists involved in the projects. The next step was to explore how these contestations were conceptualised, the aspects of commonality and difference, as well as the notable omissions. This facilitated a synthesis of the key dimensions of each contestation to inform our discussion regarding community awareness and acceptance of CCUS projects. This review concludes that each CCUS project is complex thus it is not advisable to provide best practice guidelines that will ensure particular outcomes. This systematic review shared recommendations in the literature as to how best to facilitate community engagement in relation to CCUS projects and similar place-based industrial innovation projects. These recommendations focus on the importance of providing transparency, acknowledging uncertainty and encouraging collaboration.

**Funding:** Funding: a) Grants received by: ZM b) grant agreement No 101022484 c) Name of Funder: European Union's Horizon 2020 research and innovation programme d) URL of the Funder: https://ec.europa.eu/programmes/horizon2020/en/home e) The funder had no role in the study design, data collection and analysis, decision to publish, or preparation of the manuscript. This project has received funding from the European Union's Horizon 2020 research and innovation programme under grant agreement No 101022484. The sole responsibility for the content of this publication lies with the authors. It does not necessarily represent the opinion of the European Union. Neither CINEA nor the European Commission are responsible for any use that may be made of the information contained therein.

**Competing interests:** The authors have declared that no competing interests exist.

## Introduction

The overall increase in human activities and their dependence on fossil fuel consumption has resulted in unprecedented levels of environmental degradation [1, 2]. Since the industrial revolution, scientists have recorded a steep increase in human-induced carbon dioxide emissions ($CO_2$) in the earth-atmosphere-ocean system [3] resulting in an overall global temperature rise of of 1.2 degrees Celsius [4]. This $CO_2$ increase has resulted in the rapid advancement of climate change and its associated environmental, social, and economic impacts. To address this issue a range of climate mitigation policies and strategies have been implemented stimulating the development of related technologies to mitigate $CO_2$ emissions.

Carbon capture, utilisation and storage (CCUS) technologies have, since the 1990s, received increasing attention from political, industrial and research focused stakeholders who envision that these technologies could play a crucial part in addressing climate change [5, 6]. The promise of CCUS technologies has been presented as enabling both the capture of $CO_2$ at the point of emission as well as the extraction of $CO_2$ already released into the atmosphere. Once captured, it has been projected that some $CO_2$ could be reutilised to produce different materials and what was left could be stored in underground geological formations [7].

This emphasis on CCUS as a climate mitigation option is manifested in how CCUS technologies have started to be promoted as a key component in the climate mitigation policies of the three biggest emitters, the European Union, China, and the United States. These three emitters together accounted for 52.4% of the global yearly $CO_2$ emissions in 2021[8] and 59.7% of historical emissions [9]. This policy focus has resulted in a range of initiatives to support the development of CCUS including subsidies [10], trading schemes [11] environmental directives [12], research funding [13, 14] and tax credits [15].

The realisation of these projects has however been slow to materialise. Although the component processes exist [6], CCUS as an integrated system is still a long way from contributing significantly to national and global emissions mitigation targets. For example, whilst in 2019 34Mt of $CO_2$ were captured globally, that is still only around 1% of what is estimated to be needed by 2030 if the 2-degree target is to be met [16]. Furthermore, 43% of all CCUS projects since 1995 have been cancelled or put on hold, and for larger projects, greater than 0.3 Mt $CO_2$ per year, that number increases to 78% [16].

These challenges of CCUS have led to increasing scepticism about what role it could and should play in climate mitigation policies. The potential of CCUS as a viable technology to mitigate climate change has been critiqued as the technology so far has been slow to develop [17], encompasses many risks [16, 18] and many projects use technologies such as enhanced oil recovery that increase the total carbon emissions [19]. It has also been questioned whether CCUS can be separated from capitalist and social processes that are underpinning climate change [20] and whether it will result in fossil fuel lock-in [21]. This scepticism has been expressed by the lay public concerning storage risks, uncertainties and concerns regarding the long-term efficacy of CCUS in mitigating climate change [22, 23].

The extensive debates and challenges facing CCUS technologies demonstrate climate change mitigation as a wicked problem that involves a multifaceted range of interconnected social, technical, political and economic issues [24, 25]. Individual CCUS projects themselves are complex and shaped by multiple technical, financial, political and social factors. To inform policies and debates about the future of CCUS in climate mitigation policies it is necessary to acquire a better understanding of the "wicked" nature of CCUS technologies.

One aspect of CCUS projects that has received increasing scholarly attention is how the "public" and "communities" have responded, resisted, and accepted the deployment of CCUS technologies [26–28]. Whilst the impacts and outcomes of CCUS projects may be formed by a

range of factors, the abandonment of several CCUS projects due to public opposition [29–31] has instigated the development of a significant body of literature on how social factors and community contexts influence the deployment of CCUS technologies [32–35].

Although the interest in social acceptance of CCUS technologies developed as a result of community resistance, and thus was deeply intertwined with specific community contexts and social dynamics, much research tends to focus on more general aspects of CCUS. For example, research has been conducted to map out awareness and understanding of CCUS amongst the public and key stakeholders [32, 36–40]; to examine how particular factors such as trust, economic interests and cultural worldviews shape risk and benefit perceptions [41–44]; and to probe how media framings shape perceptions [45, 46]. Whilst these studies have provided many insights into dynamics that influence lay perceptions, they do not pay sufficient attention to the multidimensional social, political, technological, and economic aspects of community acceptance of CCUS. This is particularly problematic as CCUS is a bundle of different processes and technologies with a plethora of alternative mechanisms that all have their advantages and disadvantages [18, 47–49].

In order to understand the processes that shape perceptions of CCUS, it is essential to refocus and learn from context-specific research approaches to CCUS projects. This will allow a more comprehensive understanding of the role local social context and particularities of CCUS projects have in shaping how new technologies are adopted, rejected, resisted, and enforced. Our initial scoping of the community acceptance literature identified a fragmented body of interdisciplinary work with limited coherence or cohesion around the contribution of social factors to the outcomes of CCUS projects.

To address and understand this fragmentation, we drew inspiration from the growing literature on meta-narrative systematic approaches that "*treat conflicting findings as higher-order data*" [50] (p.420), in order to explore the underlying factors that result in different findings. Meta-narrative systematic reviews can facilitate sense-making of the underlying contestations that shape a research area whilst highlighting research gaps which can be useful for researchers and policy makers seeking to comprehend a fragmented and still-emerging body of research. This article will apply a meta-narrative review to make sense of the existing research body on how communities have responded to specific CCUS projects to inform a better understanding of how communities respond to and are impacted by the increasing global policy push for the implementation of CCUS technologies. The paper recognises that this can come at the expense of being able to go into depth with the particularities of the research findings from each paper. However, given the current state of the literature on CCUS projects we believe that a meta-narrative systematic review can both help identify overarching research gaps and help illustrate how key contestations can shape understandings of social impacts and community acceptance of CCUS projects.

This review identified three key themes within the literature, namely: a) what is meant by communities, b) how is acceptance defined, and c) how are impacts perceived. These themes are explored here following a brief description of our review method. We then go on to consider the implications of our findings for the planning and implementation of international CCUS projects.

## Methods

A systematic meta-narrative review of peer-reviewed publications considering community acceptance and social impacts of CCUS projects was conducted. A meta-narrative systematic review focuses on sense-making of the research literature rather than providing a catalogue of findings [51]. It is particularly useful for examining diverse strands of research methods and

conceptualisations in order to *"expose the tensions, map the diversity and communicate the complexity"* in the field [50] (p.427). To help guide the analysis, meta-narrative reviews use the six guiding principles of pragmatism, pluralism, historicity, contestation, reflexivity, and peer review [50]. Please see the S1 File for how these principles were operationalised.

The review consisted of four phases in accordance with the meta-narrative method: search, mapping, appraisal, and synthesis phase [50]. Throughout these phases, the meta-narrative principles were implemented. This resulted in two key adjustments to this review's initial approach. The first adjustment was to focus on research that was related to site-specific projects. This was partly the result of feedback from partners working on specific CCUS projects who considered such a review more relevant for their work. It was further informed by reflecting on the initial scoping of the literature that, as mentioned above, indicated the importance of examining how CCUS projects play out in a complex local context. As the analysis developed, there was a quick realisation that research into CCUS was still comparatively new and that the interdisciplinary nature of much of the research meant that links to particular research paradigms were somewhat tenuous (see also S1 File: Reflexivity and Historicity and S1 Fig for bibliometric analysis of the CCUS literature). In view of this, a second adjustment to the review method was made to focus on areas of contestation rather than to trace out how they had been formed by particular research traditions.

During the scoping of the review, it became evident that several terms have been used for different aspects of carbon capture technologies such as Carbon Capture and Storage (CCS) and Carbon Capture Transport and Sequestration (CCTS) [6]. This diversity of terminology is also evident in the reviewed literature. What terminology to use can be a matter of extensive debate amongst researchers working on the area [52] but as it is not the purpose of this review to differentiate similarities and differences between these terminologies, CCUS is used as a catch-all term when referring to the range of technologies in the field. This does however not mean that it is not important to consider that CCUS projects can involve a range of technologies at different stages of development, ranging from pilot projects to fully commercially operating projects which all might have implications for community acceptance and social impacts. We will touch further on this issue in the results sections.

More details on the projects covered in this paper can be seen in the exclusion/inclusion criteria in the included S2 File. Here, you can find a description of the approach to each of the four phases of the review, before the results are presented and discussed.

## Search phase

In the search phase, an initial scoping review was conducted for the researchers to familiarise themselves with pertinent literature on CCUS. This allowed them to identify areas not considered in the literature and differing conceptualisations around the topic of CCUS technologies, community awareness and acceptance. The different epistemological and expert backgrounds of the researchers facilitated awareness and consideration of a diverse selection of publications. In contrast with other systematic reviews, the principle of reflexivity and pragmatism in meta-narrative reviews encourages an iterative search approach. This allowed for publications from a range of research traditions and different perspectives to be included, allowing the inclusion and exclusion criteria, and digital search strategies, to be adjusted during the search phase to reflect our engagement with the extant body of research. The search phase consisted of systematic searches using key phrases as well as forward and backwards referencing from articles. The database search terms included combinations of the terms carbon capture, storage, communities, utilisation, risks, benefits, impacts, public, and awareness. The exact search terms

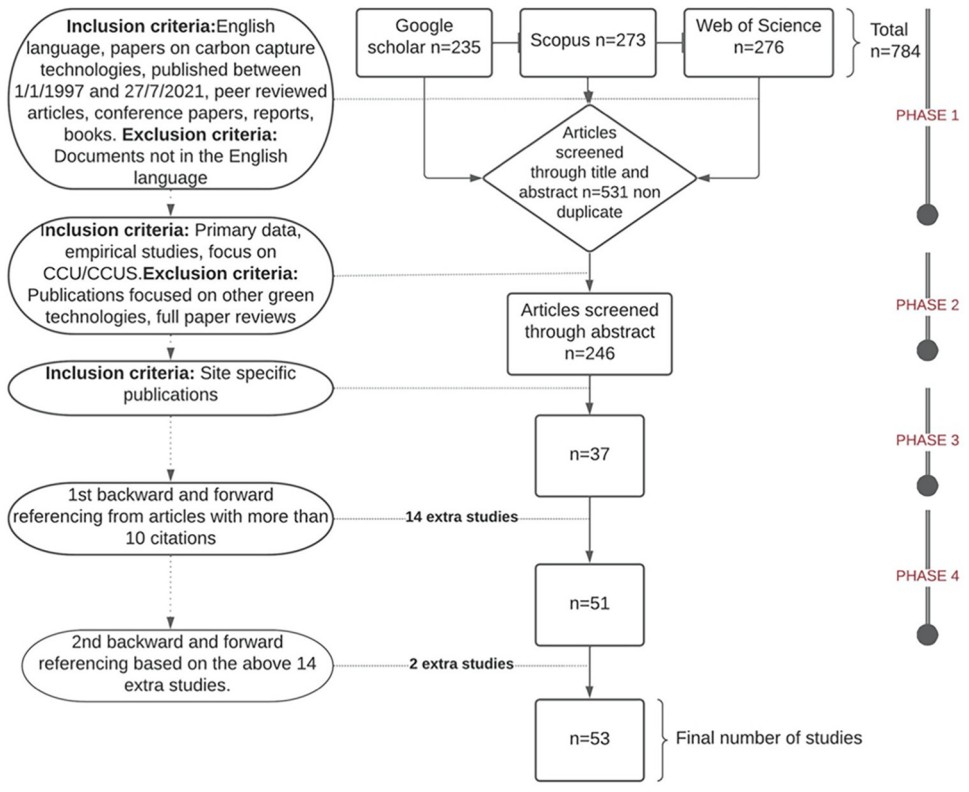

**Fig 1. Identification of sources.**

and dates can be seen in the (S2 & S3 Files). At the end of the search, we identified 53 research sources (see Fig 1)

## Mapping phase

In the mapping phase, from the total of 53 papers, only papers that had been cited more than 10 times (n = 27) in Google Scholar were thematically analysed, to identify commonalities and differences in the types of research questions, methods and theoretical frameworks used. The key findings were drawn to map out similarities and differences and started to trace some of the research limitations and areas not considered in the literature. During this phase of analysis, it was evident that acceptance, community, and impacts were concepts that had shaped the key research questions, conceptualisations, and findings. It was, therefore, decided that the review's sense-making focus should be on these three areas of contestation. In addition, the scope for greater theoretical and methodological engagement with the broader literature on community, acceptance, and impacts was identified.

## Appraisal phase

In the appraisal phase [53] the data was first extracted from the 53 papers in NVivo ((version 20) 2020). Each paper was coded for findings and discussion related to acceptance, communities and impacts as well as for location, methods, project and policy recommendations. The quality of the papers was evaluated in line with the meta-narrative approach: no priority was given to particular methodological approaches as more valid than others as different research paradigms will have different standards for what makes high-quality research. Instead, the

research was appraised based on the methodological and theoretical framework used. In the end, no papers were excluded based on a lack of quality, which may be a reflection of limiting the review to peer-reviewed papers.

### Synthesis phase

Finally, in the synthesis phase, the coded data were used to identify nine dimensions of community acceptance and CCUS demonstration sites. These dimensions were grouped under the areas of contestation where the researchers mutually agreed they related to the most, recognising that all the dimensions and areas of contestation often overlapped and were interrelated. To identify and explore these dimensions, we looked at how the research body had conceptualised them, what commonalities and differences existed between research approaches, and notable limitations and omissions in the research and evidence presented.

## Results

### Overview

53 peer-reviewed papers were assessed (see S4 File for details) reporting empirical evidence from studies on community impacts and social acceptance of CCUS-projects published between 2009 and 2021. No discernible trends were observed in the publication rates by year (see Fig 2). This may be due to several papers referring to more than one site, and some sites being more researched than others. Fig 3 shows the 53 reviewed papers and their associated CCUS study site(s) locations together with the number of papers relating to each individual

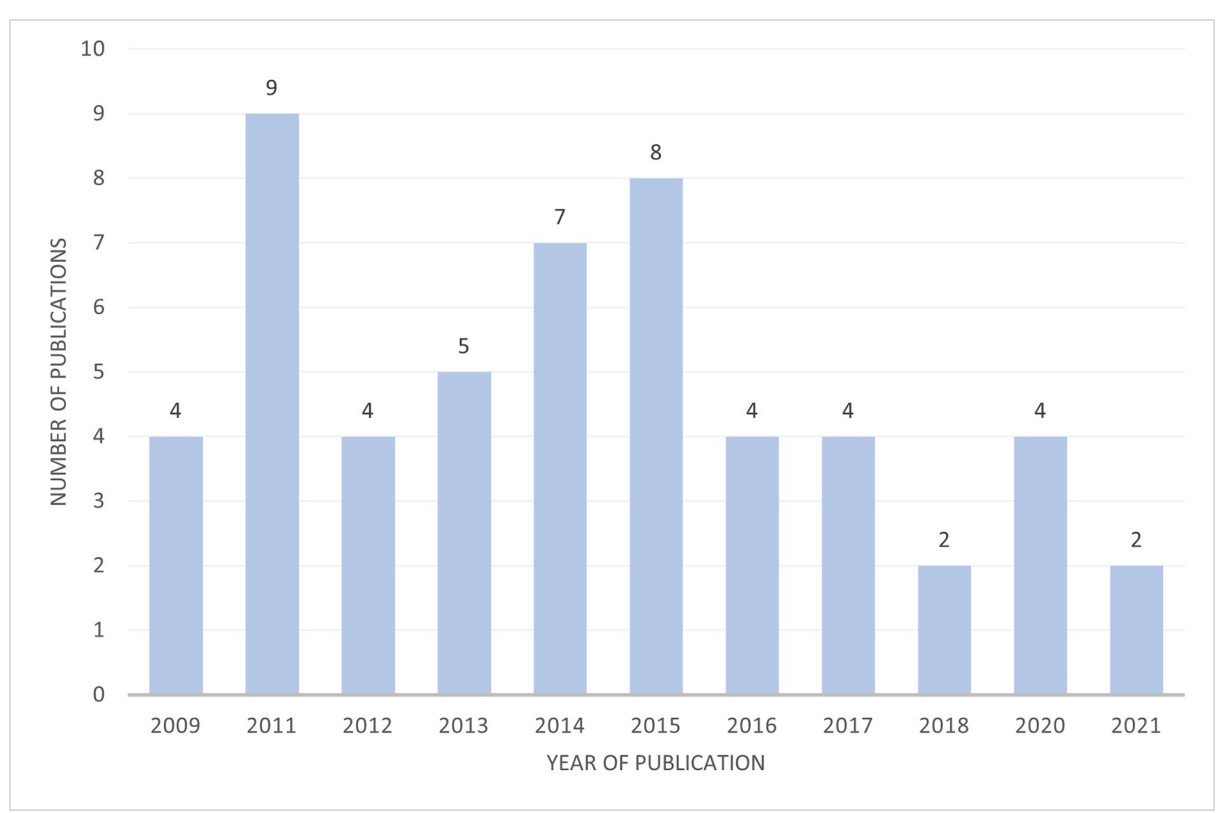

**Fig 2. CCS/CCUS site specific publication.**

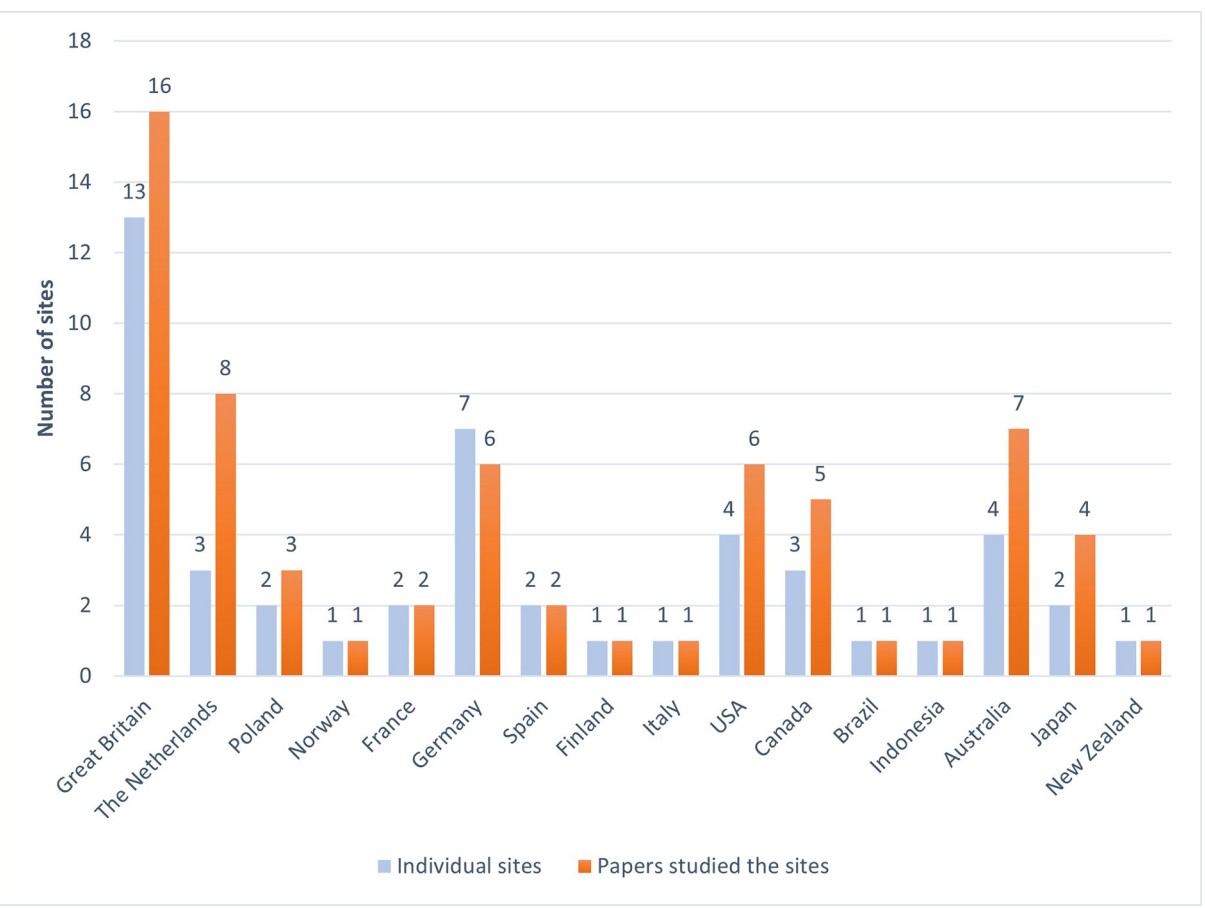

**Fig 3. Number of CCUS sites per country.**

CCUS site by country. This demonstrates the predominance of the study of Euro-American culturally situated communities. This review could not locate any studies of communities in emerging economies such as nations in China, Africa, India or the Middle East adherent to this review's inclusion criteria.

All the papers reviewed included some primary data. A small number augmented primary data with the use of secondary data. The majority of the papers, 42 (79%), reported qualitative research, seven (13%) described their methods as qualitative and quantitative, two (4%) as mixed methods and two (4%) as quantitative.

A general lack of specificity was found in the papers reviewed. For example, communities were described as local, but the exact locality was not designated. Many studies referred to research participants as community residents but did not define the boundaries of the residential area studied. In addition to this lack of specificity, a lack of research participant diversity was identified in the studies reviewed. Research examined the outcome of community engagement with only very limited evidence of community members having been involved in the research design or dissemination. Most of the research participants were adults of working age, with only rare mentions of elderly or young adults, children, or youth community members [54, 55]. Other than a study conducted in a low socio-economic status in the United States by [56] we found no direct involvement of marginalized, underrepresented and minority groups, such as faith-based, ethnic, homeless and economically disadvantaged community members.

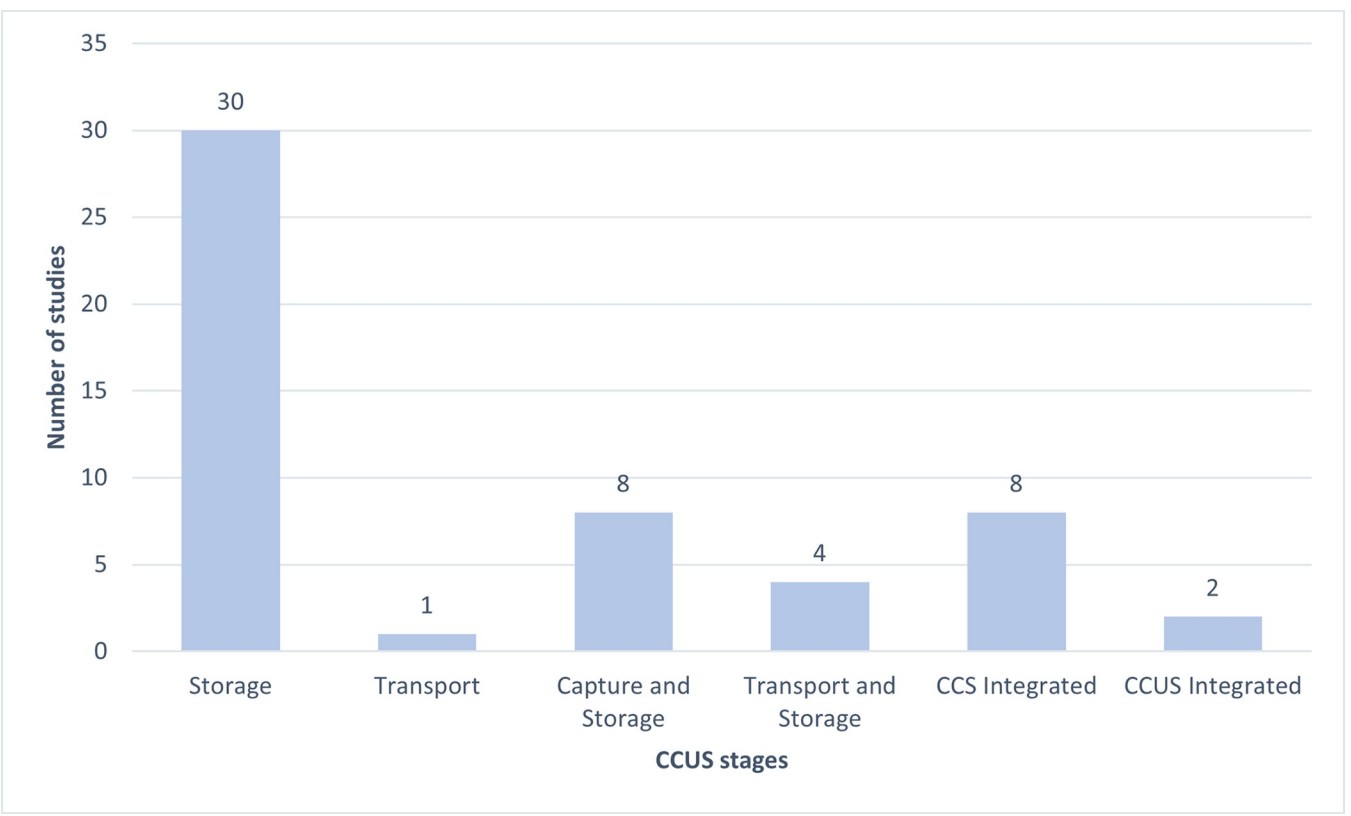

**Fig 4. Number of studies per CCUS stage.**

Slightly more than half [30] of the reviewed papers focused on the storage stage of projects, with one paper looking at the transportation stage. This focus on storage might be a reflection of how the impetus to examining community acceptance and social impacts was propelled by community opposition to the storage stage of CCUS. The rest of the papers [22] examined projects that involved more than one stage for example capture and storage [8], transport and storage [4], and fully integrated CCS projects with capture, transport, and storage [8]. Only two papers examined the utilisation stage as part of an integrated CCUS process (Fig 4). This study's criteria did not identify any studies exploring societal perceptions of Direct Air Capture projects. Regardless of what stage was the focus of the project, much of the engagement with communities explored perceptions of CCS or CCUS as an integrated process rather than focusing exclusively on the particular stages that related directly to these communities.

There was a general lack of clarity about what specific technologies were used in the CCUS projects that were explored in the papers. Some did refer to the specific technology that was tested [e.g. 30, 57, 58] however how the specific technological properties and readiness level might shape social impacts and community acceptance was left unexplored. This is problematic as CCUS can encompass a diverse set of technologies and processes with different Technological Readiness Levels (TRL) ranging from level 1 where it is only developed on a theoretical level to level 9 where the technology can be applied in an operating environment [59]. Capture technologies can include post-combustion, pre-combustion, $CO_2$ separation technology, oxy-fuel separation, and membrane separation [60, 61] and they all have a variety of physical and chemical processes and TRL levels [18, 62–64]. For the transportation stage of the captured $CO_2$, pipelines, ship, and truck transportation are the main technologies [65] with pipelines

being a mature technology and ship transportation being in the demonstration and large prototype stage [63]. As with the other stages of CCUS, there are different TRL levels in the development of $CO_2$ products. As an example, according to [66] urea production has reached a commercial (TRL9) level where as formic acid and calcium carbonate are at lower TRL levels, 6 and 7 respectively. Technologies related to storage of $CO_2$ can involve saline formations, depleted oil and gas reservoirs as well as enhanced oil recovery processes [61] with TRLs ranging from 9 for enhanced oil recovery and storage in saline formations to earlier stages of development for other storage options such as mineral and ocean storage [18, 67, 68].

Given the wide variety of technologies involved in CCUS and considering their different TRLs, it raises the question about how this might relate to community acceptance and social impact. Although the link between technology readiness levels and social acceptance by communities is still underexplored there is research that in work settings indicates a link between acceptance of technology and technological development levels [69, 70]. Furthermore, studies also indicate that consumer acceptance of new green products are linked to technological readiness levels [71, 72]. However, in the reviewed papers these aspects were not explored leaving considerable room for further research.

## Contestations

This meta-narrative review identified acceptance, community, and impact as three key areas of contestation in the research literature. This review suggests that how each of the papers approached and used these terms shaped how they conceptualised the problem, the type of research questions they asked, the methods they used, the findings they arrived at and the recommendations they promoted. Within each of these three main areas of contestation, a further three dimensions were identified (giving nine dimensions in total) illustrative of the underlying dynamics that had shaped understandings of acceptance, community, and impacts, as summarised in Fig 5. The following section presents our considerations of each of these nine dimensions of the three contestations: acceptance, communities and impacts.

## Acceptance

Whilst community and social acceptance of CCUS technologies were generally considered to be important for the successful implementation of CCUS projects throughout the papers reviewed, a wide range of understandings around what is meant by acceptance, how it should be achieved, and what purpose it should play in the wider project was identified. How '*successful acceptance*' was conceptualised had implications for CCUS project evaluation, the type of engagement practices that might be recommended, and the social factors deemed to be important in achieving those goals. To illustrate these tensions, three key dimensions of acceptance were examined related to contestations on how acceptance is conceptualised, the contribution of engagement to achieving acceptance, and the role social knowledge plays in the project.

**Understandings of acceptance.** The first key dimension of acceptance is how it is conceptualised and understood as a term. One of the most common understandings of acceptance in the reviewed papers was as passive resistance indicated by a lack of publicly visible resistance. For example, the absence of public protests was interpreted as an indication that a project was accepted "*or at least quietly tolerated*" [73] (p.6235) and when surveys found a willingness to sign petitions against a proposed $CO_2$ storage project this was seen as a sign that the community did not accept the deployment of CCUS [27]. Acceptance was sometimes understood as consensus amongst established groups, for example where the interests of relevant stakeholders within a particular policy framework had aligned and thereby eliminated any visible opposition [57].

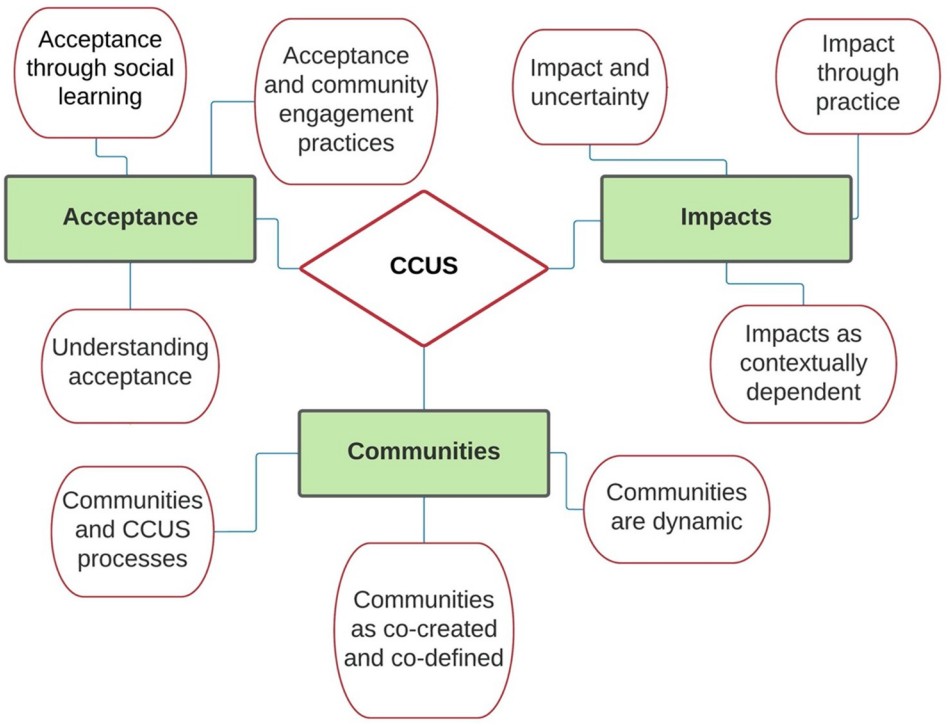

**Fig 5. Review themes and sub-themes.**

This implicit assumption of acceptance is interesting. In several of the reviewed papers what is meant by "*observed acceptance and resistance*" [74] (p.403) is not clearly detailed. Whilst local, social, and public acceptance may be mentioned several times [e.g. 75], it is not always clear why a lack of public resistance is considered to indicate social acceptance. This lack of clarity might be due to the policy-focused context that these papers emerge from. Within the policy context of CCUS, and other renewable energy technologies, acceptance is often conceptualised as a top-down process whereby energy infrastructure projects are "*gifted*" to local communities who are expected to passively accept and tolerate them [76]. Seen from this perspective the purpose of community acceptability research is to devise engagement strategies and practices that can ensure passive community tolerance of the project.

Although most of the literature approached acceptance as a lack of resistance, there were a few studies that contested this approach. One study of a project in Otway, Australia [77] found that some local communities could not access proper information about the project and did not have the resources to negotiate and engage with the project partners. This resulted in passive acceptance from the community as they did not have the capabilities to make their voices heard. Consequently, even when the project started to disrupt their businesses and cause some personal injuries, it did not result in open protest despite increasing dissatisfaction with the project [77]. Similar issues were found in a piece of research that compared how two communities near Sacramento, California with different material and social resources reacted to a CCUS project [56]. They found that the community that was socially and economically disadvantaged had the same concerns about the potential risk of the projects as the more affluent community, but the disadvantaged did not expect to have their voices heard when addressing these issues due to a sense of disempowerment [56].

Some papers found it important to consider how to move beyond passive tolerance and acceptance of projects. One suggested approach was to focus on capacity building of the local communities so that they were able to fully access all relevant information about the project and so that they were supported in their efforts to influence how the project is carried out and negotiated [77]. Another example of how this could be done was the ZeroGen project where a partnership consultation process was established in the planning phase and legal and technical resources were provided to the involved indigenous communities to increase their capacity to influence the project. This consultative process consequently led to a sense of partnership between the indigenous communities and the project team. Although there could still be disagreements the consultation was conducted in an atmosphere of trust [78]. These findings resonate with other research that indicated that communities that feel empowered by their capacity to influence projects and mitigate any potential risks are more likely to accept pilot projects [29, 56, 79].

Two of the papers questioned whether community acceptance was a crucial factor in deciding project outcomes. They argued that whilst the abandoned Barendrecht project is often taken as a case study to illustrate the importance of gaining community acceptance, it was rather the conflicting views amongst key policy stakeholders at the national level combined with a lack of understanding of roles and responsibilities amongst market actors that resulted in the project being terminated [80, 81].

This current review found that there was scope to engage more critically with how acceptance is conceptualised and understood by the project partners. Looking at the wider policy discourse on climate mitigation and adaptation, the concept of communities as passive recipients of climate mitigation and adaptation initiatives goes against many United Nations (UN) agreements, ranging from the Aarhus Convention that guarantees the right to information, participation, and justice in relation to environmental decision making [82] to the more recent Paris Agreement that emphasises the importance of participatory approaches, local knowledge systems and the rights of communities and vulnerable groups [83].

This emphasis on community-based and participatory research approaches also ties in with findings from the wider literature on climate change mitigation and adaptation that has highlighted some of the limitations of top-down solutions [84, 85]. As climate change was initially mainly conceptualised to be a global problem much of the early approaches to tackling the issues relied on global models that could not take account of local complexities [86]. This resulted in projects that implemented solutions that were ill-suited for the particular social and cultural context [87, 88] and that produced mixed results with unexpected adverse political, social, and economic consequences [89]. As a way to try to overcome these issues it has been suggested that more participatory and community-based approaches have the potential to empower communities [90–93], widen participation [94, 95], linking across scales [85, 96], and better utilise local knowledge [85, 93, 97].

This approach does not mean that participatory methods are a panacea that can overcome all the complexities of climate mitigation and participation initiatives. There are still important questions that need to be examined about the efficiency, impact, and sustainability of participatory methods [98–100] and it is still necessary to acquire a better understanding of the interlinkages between community processes and climate change mitigation and adaptation initiatives [85, 101]. Nonetheless, a participatory process that seeks active community acceptance and support could be argued to be crucial to ensuring the sustainability of climate change mitigation and adaptation. Also, although many of these lessons emerge from within the literature on climate adaptation, there is still scope for engaging more with these findings when it comes to understanding the relationship between acceptance and climate change mitigation initiatives such as CCUS projects.

Another issue that could be considered more thoroughly is that whilst a lot of the focus on social acceptance and communities emerges from concerns about resistance to CCUS projects, resistance is not necessarily problematic in itself. Several scholars have suggested that disagreement has great value in environmental issues and should be welcomed and embraced as it promotes pluralism in opinions [102, 103]. Ho [104] suggested that for the disagreement to be fruitful there needs to be mutual respectful discussions amongst all actors, as found in [103].

Another strand of research that could be fruitful to engage with is the literature that examines how social acceptance is a complex and multilevel construct that is important for both societies and businesses. Social acceptance of an innovation is an amalgamation of three separate levels of acceptance including socio-political, market and community acceptance [105]. Socio-political acceptance refers to the acceptance or rejection of technologies and policies by social actors such as policy makers and the lay public [80]. On the other hand, market acceptance considers the interactions between consumers and technology investors both in a competitive and collaborative context [106]. The third level of social acceptance is community acceptance referring to a project's acceptance from local stakeholders including residents and the local authorities [105]. These three levels of acceptance can be argued to interlink and influence the outcomes of the CCUS project [80, 106].

To conclude, although acceptance is mostly approached as a lack of resistance in the literature, this kind of approach has been contested by a minority of studies that emphasise the importance of community capacity building to enable a more active form of acceptance and the importance of critically examining in what circumstances community acceptance becomes important. This review suggests that learning from the wider research on participatory methods, resistance and disagreement, as well as approaches to socio-political and market acceptance will enable a more robust understanding of the issues around how acceptance is approached as one aspect of meaningful community engagement practices.

**Acceptance and engagement practices.** In the literature, methods of engaging with communities and stakeholders were commonly understood to be crucial in relation to the implementation of CCUS technologies and issues with the engagement processes were often found to be at least one of the explanatory factors behind why projects might be abandoned [e.g. 27, 29, 57, 73, 75, 107]. However, some contestation about what the aim and purpose of the engagement practices should be was found in the literature.

The most common approach to engagement practices was that they should help facilitate and improve the likelihood of community acceptance of the CCUS technologies [e.g. 29, 108–110]. From this research perspective, one important element in shaping the engagement process was the timing of engagement. A lack of early engagement might be considered by some communities as an indication that their concerns were not being taken seriously, which might in turn increase resentment towards the project [29, 111]. As far as possible, communities should therefore be involved early in the process, allowing time for them to digest the information [58]. Preferably the community should be engaged before any final decisions are made in terms of whether to implement the project at all, although that can be practically difficult to achieve [58].

For a few papers, the purpose of engagement was not only important to facilitate acceptance but could also play a part in promoting wider social justice issues. In the literature, three interrelated notions of justice were found and used to enhance the community engagement project, namely: procedural, distributional, and epistemic justice.

Procedural justice refers to whether the decision-making process is fair, transparent, and participatory. This could refer to the timing of the community engagement as mentioned above and related to past experiences. For communities that had been through a consultation process for other industrial and environmental projects, there could be an expectation that the

engagement process would be merely procedural [112]. This could then result in a negative circle where a perceived lack of procedural fairness resulted in reduced willingness to participate in community engagement initiatives [107].

Distributional justice relates to how fairly and equitable the benefits and risks of CCUS projects are distributed across different communities. In the literature, there were some cases where the issues of distributional fairness were found to have an impact on how communities perceived projects. This concern was tied in with spatial dimensions. For example in connection with a project in Queensland where the $CO_2$ would be used for enhanced water recovery, members of the communities raised concerns about how the unequal spatial distribution of risk in terms of $CO_2$ leakage and the benefits of improved water levels would benefit some whilst disadvantaging others [30]. Similar concerns were found in a study looking at community responses to new energy projects in Ontario, including a CCS project in Alberta. Here the rural Ontario communities considered that they had to shoulder most of the risks of new energy developments whilst urban centres would receive most of the benefits [107]. Concerns about project motives were also found to contribute to a sense of unfairness in how the benefits and risks were understood. For example, there could be scepticism about whether the project was about addressing $CO_2$ emissions or facilitating capital accumulation for project partners [107]. This focus on distributional justice informs the policy focus on '*just transitions*' that seek to ensure that communities and workers are not left behind or disadvantaged in the transition away from carbon-intensive industries [113].

Epistemic justice goes beyond distributional and procedural concerns to consider how people may feel marginalised and excluded when it comes to shaping the types of questions that are being asked and accessing the knowledge that is being created in CCUS projects [114]. Community participants were sometimes found to express notions of epistemic injustice if the CCUS project was seen as being part of a bigger policy where it was assumed that widespread CCUS deployment would and should happen [114]. For example, in the Barendrecht project, questions about the necessity of CCUS and potential alternatives were deemed irrelevant and suppressed during the community engagement [115]. This led to increasing public resistance against the projects which made the project politically problematic and ended with the abandonment of the project [115].

Whilst these notions of procedural, distributional and epistemic justice highlight important factors that can shape community impacts and acceptance, few examples of how to overcome these issues were found. One approach to addressing them was by seeking to use particular forms of community engagements such as "*deliberate engagement process*" and "*deliberative workshops*" [116, 117]. Such approaches would focus on active listening, the co-creation of knowledge, and the inclusion of diverse views from communities and stakeholders with the hope that, by seeking to foster collaboration, power imbalances between the local community and project stakeholders might be addressed [116, 117].

Although using more collaborative engagement processes is promising in terms of addressing some issues of injustice, it is still unclear to what extent they would be able to deal with procedural, distributional, and epistemic justice concerns. In the wider literature on climate mitigation and adaptation initiatives, there has been criticism of the notion that project practices, no matter how well-intended and how much they seek to be participatory, can end up being symbolic, ignoring structural inequalities and the ways communities may be shaped and influenced by political, social, economic, and environmental factors beyond their control [85, 87, 118–120].

To sum up, although there was agreement in most papers that the practices of engaging communities could play a role in shaping acceptance, the scope and aim of community engagement practices differed in the literature. Whilst some only focused on the timings and

content of the community engagement practices [e.g. 27, 121, 122] other papers conceptualised the scope in much wider justice terms [e.g. 30, 107, 113, 114]. It was however not always clear to what extent community engagement practices in themselves could address the range of issues and concerns that shaped community acceptance.

**Acceptance and knowledge.**   Another key dimension identified in this review was how the link between knowledge and acceptance was understood in the literature. One conceptualisation of this relationship was that community and stakeholder perception of CCUS projects was shaped by a lack of awareness about the CCUS technologies that would be implemented in the project. This could then in turn lead to attitudes being shaped by a range of beliefs about CCUS that did not correspond with experts' views. From this perspective, the best way to facilitate acceptance was to identify what patterns of awareness and beliefs people had about CCUS projects [27] and to consider how best to inform people about the technology [115].

However, there were a few papers that examined how the involvement of communities in the CCUS process could be beneficial in terms of contributing to what was termed as the 'social learning' of CCUS projects. Projects are often framed as part of a technical process, where scientific methods are utilised to establish objective facts that are to be communicated to the public and other stakeholders. However, the social learning approaches tie in with findings from social science that have questioned many of these assumptions [123, 124]. CCUS projects, like other technological developments, are characterised by many uncertainties [58] and are "*a process that is not just about learning technical facts, but also learning about other aspects of the technologies integration into society*" [124] (p.6249).

A paper by [57] illustrates how a broader social framing of project scope could facilitate social learning. By examining three projects in the United States (US), United Kingdom (UK) and Japan they found that projects that moved beyond technological learning objectives also enabled lessons to be learnt regarding wider social factors as they relate to CCUS projects. For example, one paper found that the Yubari project in Japan had a technocratic focus as "*the justification of the project, the overall goals and results and the project's relationship with society have been presented almost exclusively on technical and economic grounds*" [57] (p.299). This focus on establishing technical and scientific facts resulted in minimal community engagement with consequently little chance to learn about wider social aspects of CCUS technologies. In contrast, the FutureGen (US) and Longannet (UK) projects sought to look at broader social aspects such as acceptance, education and legal regulations. This allowed more comprehensive social learning lessons to be achieved, although there was still much scope for wider community engagement in these projects [57].

Similar issues can be found in the paper by [54] where they reflect on their own journey as dissemination experts in the CCUS field. They started with a narrow framing of the issues that resulted in a top-down approach that assumed that they as experts knew what the problem and its solutions were and that the "public" was nothing but a barrier lacking insight and understanding. Yet as they became more involved in projects, they realised that this approach did not "*recognize all the complex socio-cultural aspects that support innovation and problem-solving at the social level*" [54] (p.4839).

A few practical suggestions were identified for how to open up space for social learning that moves beyond an exclusive focus on technical expertise. One suggestion was to try to move towards approaches based on mutual learning patterns [54]. To exemplify the benefits of this approach the researchers engaged with school children aged 9–10 and illustrated how this engagement had helped the researchers clarify the concepts they used whilst illuminating the wider social context of CCUS [54]. Another suggested approach was to classify CCUS projects as "*unstructured problems*" [29] (p.6380). This approach would seek to open up for wider questions about whether CCUS is useful and how it compares to other technologies. It would also

seek to create opportunities for more inclusive dialogue between stakeholders and communities with different beliefs and values [29].

Alongside these different understandings of knowledge, it is important to consider how and by whom knowledge is produced and used as well as how there is a relationship between power and knowledge [125–127]. In his writings, Foucault criticizes the paradigm that knowledge should only come from institutions-such as universities- with sciences holding the power of truth [128]. In an attempt to address that issue, scholars have looked at participatory action research (PAR) as a platform to overcome the domination of knowledge associated with conventional research methodologies [125]. PAR allows for a more democratic process of knowledge development through the voices and experiences of local people [129, 130].

Pluralism of opinions can also be argued to be essential for democratic processes, as, through the expression of diverse voices and social interactions, societal conflicts can become more apparent and accessible to everyone [131, 132]. Furthermore, communities are important social spaces to in themselves, as they can play an important role in peoples' knowledge development through social learning. Learning is a social process that takes place inside a social system [133], thus it is important to look at the implications it has on knowledge development amongst the different actors involved in CCUS projects.

The role of knowledge in environmental governance has furthermore also been discussed and explored by several environmental scholars [134, 135]. There is increasing recognition of the importance of local communities and indigenous people and their associated knowledge systems to climate adaptation policy [119, 127, 136, 137]. Indeed, both scholars and the Intergovernmental Panel on Climate Change (IPCC) have established the importance of knowledge generated by diverse knowledge systems in addressing climate change [127, 136, 137]. Although the importance of indigenous people has been researched and acknowledged, the necessary policies and strategies are not in place to incorporate that knowledge in climate change policy development [127]. This was also evident in our review that the experiences and knowledge of more marginalised communities were not represented or recognised nor used to build up more diverse ways of knowing and understanding the implications of CCUS technologies.

## Communities

As communities are considered to play a central part in the social acceptance of CCUS technologies, it is critical to examine what is meant by *"communities"*. Terms such as 'community' and 'stakeholders' were frequently used but rarely specified or defined. Given the importance placed on community acceptance, it was perhaps surprising that definitions of communities and how they might be represented were seldomly discussed in-depth, highlighting an area to be further developed in the CCUS project literature.

**Defining and representing communities.** This review identified a range of approaches to defining communities. Some studies approached communities in terms of geographical boundaries centred on geographical locations [27, 73, 77], whilst others based their definition of communities and stakeholders around social interests in the project [78, 80, 138]. Some used more ad hoc approaches such as equating the readership of the local newsletter with the community [121].

There were also several different ways to conceptualise who represented these communities. One common approach was to consider stakeholders associated with local government, Non-governmental organisations (NGOs), academia, industries, and expert communities as representatives of their respective communities [54, 80, 81]. Another approach referred to groups of farmers, local residents, and landholders as being representative of the local communities

[117, 122]. Others saw media as simultaneously reflecting and influencing community attitudes, centring part of their analysis on media discourses to reflect community concerns [73, 75, 117].

Most studies provided no extensive description of how the communities had been defined nor why they had chosen particular ways of representing these communities. This produced analytical ambiguity. For example, in a study conducted in Priddis, Canada, participants were grouped as *"1) the general population of the area and 2) those who had a particular stake in the development or played a clear role in the opposition of the project"* [139] (p.190). It is not clear here whether a participant opposed to the project (Group 2) was also a member of the general population (Group 1) as all the interviewees were residents of the area too.

This issue of definition hints at the complexity involved in studying communities. Notions of communities as simple, natural, discreet and static parts of the social world have been criticised for being based on unstated assumptions rather than a critical examination of what communities are in all their complexity [140]. How communities are formed, contested, understood and experienced relates to a range of local and global social, economic, material, and political processes [141, 142]. Whilst communities are often studied and conceptualised as a single unit tied to place or interests, it has been highlighted that actors interrelate across networks that often transcend these imagined community boundaries [143]. Furthermore, over-emphasis on communities of place and interest fails to attend to ways in which micro-level interactions shape and rework these communities [144]. This is challenging for CCUS projects as environmental issues tend to interlink and move across the scales of historically constituted imagined communities [85, 145, 146]. How communities are defined, understood, and approached therefore also has implications for how community impacts and acceptance is made sense of in the reviewed literature.

**Locating communities.** Our review identified a specific issue related to how the link between CCUS processes and the location of the communities was conceptualised. CCUS technologies refer to clusters of interconnected industrial processes and developments that can take place across vast geographical regions, interconnecting with wider environmental, social, and economic processes. This review was also interested in what part of these industrial processes is focused upon when researchers locate and identify relevant communities to engage with. The review found that the majority of the research focuses on the connection between community acceptance and the storage of $CO_2$ whether that was in projects that focused exclusively on the storage stage or whether it was in connection with studies that also explored the carbon capture, transportation, and/or utilisation stage of CCUS. (See Fig 4 in the results section).

This focus on the storage aspect of CCUS was often justified by referring to how many of the most well-known and impactful examples of community resistance has happened around suggested storage sites [e.g. 74, 80, 81]. These approaches were linked to a goal-rational framework where communities are seen as risks and barriers that can disrupt CCUS projects at different stages [75]. Papers that moved beyond the storage perspective included studies by [55, 147, 148] who explored the capture, utilisation and transportation aspect of CCUS. As mentioned in the discussion on acceptance, this fails to acknowledge justice and human rights issues of climate change mitigation initiatives and the potential for community participation to enrich social learning and project outcomes [124].

Although the cases that have received most research attention have been storage projects abandoned due to visible public protest, this should not be assumed as an accurate reflection of all CCUS projects, nor does it mean that this pattern will replicate for all projects as some community responses will be contextual and/or determined by the conduct of the particular project. As CCUS covers a range of integrated and connected suites of processes, examining

communities solely connected to one aspect of this technology (e.g., storage) may limit our understanding of underexplored but crucial aspects of community acceptance as it relates to any CCUS project.

As discussed, communities can be conceptualised in a range of different ways, and it can therefore make sense to define communities dependent on the particular research context and framework. Consequently, research studies should utilise concepts of what a community is as determined by the place and particularities of that project, rather than on how this has been done in other projects with different sets of characteristics. For some CCUS projects where the storage element is central to the project aims it might make sense to focus on local communities near storage sites. Although there are no comprehensive databases that encompass pilot and demonstration sites [16] commercially operating CCUS facilities have so far been limited to capture and storage [149]. However, many of the ongoing and upcoming demonstration and pilot projects involve the utilisation element and one database estimates that 139 ongoing or planned projects involve a utilisation stage whereas the same number for projects that focuses solely on the capture and storage part is estimated to be 192 [150]. Although capture and storage continue to be prevalent in CCUS projects there are also indications that the utilisation element could become more significant. It is therefore important to pay attention to how these evolving technologies might also change how relevant communities are conceptualised and located as well as how engagement practices are shaped.

**Knowing communities.** Who has the knowledge to define, delineate and understand communities was another dimension that shaped the research literature. This dimension has wider procedural, distributional, and epistemic justice consequences as definitions of communities can have impacts on who is consulted in the process and how benefits and risks are distributed. In much of the research reviewed, it was the sole purview of the researcher to define, delineate and determine the project's relevant communities, suggesting a view of communities as distinct, unchanging, and clearly visible entities [73].

There were however some other ways of understanding how to explore communities that move beyond the views of the researcher. For example, focus groups conducted at two potential CCUS project sites in England left it open for the participants to deliberate on how to define relevant communities that might be impacted by a potential CCUS project [108]. Within these discussions, complex understandings emerged of the relationship between specific parts of the CCUS technologies and who the relevant communities and stakeholders might be. So for example the participants considered that the local place-based community was important when it came to the capture of $CO_2$ whereas when it came to offshore storage the community were conceptualised at the national level [108].

A similar approach was found in a paper focusing on how CCUS project experts approached the issues around engaging with and acquiring a social license from relevant communities. Focus groups amongst experts uncovered the complexity involved in defining and locating the "right" kind of community to engage with. For example, the problematic nature of locating "local communities" when the storage site was located in far-away offshore storage sites was highlighted. Interestingly, a more utilitarian and goal-oriented approach was evident amongst the CCUS project experts in the study as they considered the ideal local community to be one that was sparsely populated as that would make community engagement more manageable [151].

Who gets to define and know the communities ties into concerns about epistemic justice, what knowledge-making role communities should have in CCUS projects, and how best to engage with communities. This question of who gets to define communities is interconnected with questions of definition and representation. As PAR aims to democratize research and give voice to the local people [152], it might be that PAR participants are the ones that define

what community they are part of. Even if approaches draw on the knowledge of stakeholders and communities in defining relevant communities there are still questions about what communities and stakeholders' knowledge is sought out. Given the highly context-dependent and ever-transforming nature of communities as they relate to CCUS projects we cannot recommend a single correct answer to these questions. There will be many unknowns and complexities regarding who the communities might comprise. We argue that it is important to acknowledge and examine this complexity more thoroughly to achieve a better understanding of the nuances of interrelated communities.

## Impacts

Impacts emerged as one of the key areas of contestation that were seen as shaping the wider dynamics of community acceptance of CCUS projects. Although CCUS is often framed as a necessary and beneficial technology, like any other technology there can be potential risks that need to be taken into consideration when evaluating the impact [153, 154]. Risk can however also have positive connotations and be associated with potential rewards, benefits and opportunities as often seen within management and businesses [155]. Although risks and benefits are sometimes approached separately they can be inversely related to each other, and interlink with different technological, health, business, financial, and environmental aspects of our daily lives [156–159].

How risk and benefits are perceived and contested can be considered to play an important role in shaping people's attitudes to technology, such as nuclear energy [160], renewable energy [161], as well as carbon capture, utilisation and storage technologies [23]. It is therefore important to examine how impacts are conceptualised, made sense of, and dealt with in practice in CCUS projects.

**Delineating impactful factors.** A key dimension shaping how impacts were conceptualised in the papers reviewed concerned what kind of factors and dynamics might shape people's perspectives. All papers recognised that social factors played a role in people's understandings and perceptions of impacts. There were however some differences in the role these social factors were seen to play and how they related to other physical factors.

Some papers took the position the impacts, and in particular, the risks, of CCUS technology projects were material properties that can be accurately estimated through technical risk assessments [110] and delineated from social perceptions of risk [162]. From this perspective, social factors can shape how these risks are perceived and made sense of, but they do not relate to the "objective" realities of these risks.

Other approaches focus on the social construction of CCUS technologies and their impacts [29]. In these approaches, impacts are inherently tied to different social factors and practices such as culturally embedded narratives [117] identity, history, social trust and economic context [163]. It is therefore not possible to separate any objective material facts as these facts are inherently tied to social dynamics.

These contestations about how to understand risk relates to wider debates where some argue that understanding risk is mainly an "objective" exercise where it is possible to calculate the probability for an action to take place and estimate the uncertainties around the consequences of that action [164, 165]. Yet others argue that risks and benefits cannot be narrowed down to objectivist estimates that can then be communicated to the unknowing public. Instead, it is important to examine what factors, dynamics and contestations shape risk and benefit perceptions amongst different communities and stakeholders [166, 167].

Regardless of how social and physical factors were delineated, most papers sought to identify particular important factors that had shaped people's perspectives on a particular CCUS

project. These factors tended to relate to four key aspects of risk: 1) financial; 2) environmental; 3) health; and 4) socio-cultural. Often the perceived risk of CCUS projects stemmed from uncertainty regarding the technology used, lack of trust in the project owner, lack of technical knowledge, previous experiences with similar projects and technologies, and the lack of involvement in decision making [75, 77, 116, 151].

How these different factors ended up shaping risk and benefit perceptions seemed to depend on the specific project context. Across the reviewed literature it was common for certain issues (e.g., financial and environmental) to be perceived both as a risk and as a benefit by different actors. This distinction often depended on the role of the respondent. The environmental dimension of risks and benefits is a good example of this distinction. One of the environmental benefits associated with CCUS is as a potential means of climate change mitigation as suggested by some participants [55, 74, 77]. At the same time, the environmental dimension was perceived as a risk associated with groundwater and soil contamination, marine pollution, natural disasters, and impacts on biodiversity [74, 79, 109, 163, 168].

Another example of differing perceptions of risks and benefits is how financial aspects are often perceived as a risk to the implementation of large-scale CCUS projects by corporations and employees [57, 147, 162]. Conversely, the potential economic transformation such large scale projects can bring to an area might also be considered one of the main benefits amongst communities [108]. More specific factors might elicit different interpretations. In some cases, CCUS projects were seen as potentially damaging to local tourism and real estate values [169], whereas in other project site locations with different local and social characteristics participants believed that CCUS would attract tourists and financial benefits for the community [73].

The characteristics of CCUS technologies may also influence people's perception of risk and benefits. As a high impact technology [170], characterised by spatially wide-reaching networks and potential social and environmental impacts across multiple scales, the risks and benefits profile presents some particular challenges as they can often be unequally distributed [171]. These concerns link with concerns regarding CCUS and distributional and procedural justice. Some communities might feel they are not being involved in the decision-making process and they are being exposed to most of the risks whilst others reap the benefits [30, 107, 112].

In summary, whilst our findings identified a range of different types of factors that shape perceptions about impacts, be they economic, environmental, social, or health-related, how these factors play out in other specific contexts cannot be predetermined.

**Impacts and uncertainty.** How to deal with the uncertainties about the potential impacts, whether seen as benefits or risks, was another key dimension shaping research approaches. That different factors can be interpreted in multiple ways only covers part of the complexity involved in determining risks and benefits in CCUS projects. Examining the wider research on risk illustrates cultural structures, identities and lived experiences can influence risk perceptions [172, 173]. This tension again speaks to the difficulty of examining risk as an abstract concept that is shaped by different social factors and the notion of risk as a physical reality with material properties [165].

In CCUS projects, both risks and benefits are dynamic constructs that can change throughout a project [78, 109]. There are different risk and benefit associated dynamics for the various dimensions of CCUS, i.e., capture, transportation, utilisation and storage [23, 174–176]. Furthermore, it can be difficult to fully predict how the specific technologies interact with complex environmental processes. For example, the risk of $CO_2$ leakage [177], could have multiple implications for human health, biodiversity, and groundwater contamination [178, 179]. Whilst the focus might be on the immediate risk (leakage), the implications can be multiple, and their risk profile can be difficult to evaluate in isolation.

The way the papers we reviewed approached this complexity differed. Some took a more technocratic view considering that given the potential risks and benefits is such a complex phenomenon it would be better to leave it to the technical, scientific, and social experts to deal with [110, 180, 181]. From this perspective, it was recognised that communities' views on the impacts of CCUS projects were shaped in interaction with complex social-cultural dynamics [75, 180]. However, these social-cultural dynamics were understood as an element that needed to be considered when designing engagement practices in ways that would avoid misunderstandings regarding the underlying risk factors [182]. A two-way communication approach with the communities was recommended, to build trust in the expert's ability to manage the impacts of the project [180].

Other approaches put greater weight on the communities' ability to make sense of and contribute to complex understandings of the potential impacts of CCUS projects. For example, one study found that lay people provided with some basic information about the technology can comprehend and address complex issues associated with the risks and benefits of CCUS projects [182]. In some studies, communities illustrated this complex understanding of the issue by recognising that there would always be risk involved with any kind of project [114, 162]. However, community recognition of the potential risks of projects did not necessarily result in a rejection of the project. Rather it raised further questioning about whether the project team recognised the limitations of their expert knowledge and had made adequate preparations to address any unexpected outcomes [58]. When experts and laypeople participated together in citizens' panels it was found that the two groups' risk perceptions converged after the panel meetings had taken place, illustrating how these meetings can also help develop the experts' knowledge about the potential impacts of projects [162].

**Challenging impacts.**   Many papers recognised that CCUS projects were often contested and that this related to how the potential impacts of the project were perceived. Whilst there were some examples of collaboration between experts and local communities, local communities often contested the information provided by the CCUS project partners. The potential risks, dangers and benefits were often found to be the main points of disagreement amongst companies and local community members. However, within the literature, there were different understandings of why these conflicts emerged.

The most common approach was that these conflicts could be mitigated through communication and public engagement, raising the question of how best to communicate and engage with the community to alleviate these tensions. For example [183] reported that amongst five different projects they reviewed, the projects that involved early communication and outreach were the ones that experience the least resistance against them. Often that success was due to trust-building amongst companies and local communities, as the local public trusted that the project owners would take all the required risk and safety measures to ensure the smooth operation of the facility [115, 183]. On the other hand, companies often failed to establish any trust with the impacted communities, and what these companies reported as risk and safety measures were seen as inadequate by the public [30, 56, 112].

This emphasis on the importance of communicating and engaging constructively with communities when dealing with risk links to particular risk perspectives that situate risk in psychological characteristics. Within this tradition, risk perception is how we understand and acknowledge the risk. Risk perception has been studied both from a cognitive and an affective perspective [184–186], as well as a dual approach process where both cognitive and affective assessments are deliberated [187]. Cognitive risk perceptions refer to logical and analytical decision making, whereas an affective risk perception is associated with emotional and heuristic-based decisions [188]. In extension [171], describe technological risk perception as the process where a person takes into account the "*physical signals and/or information about potential*

*hazards and risks associated with a technology and the formation of a judgment about serious-ness, likelihood, and acceptability of this technology.*" (p.293). That judgment is a combination of knowledge, values and feelings towards the technological risk [171].

This emphasis on communication and engagement regarding project risk was however not uncontested. A number of papers highlighted how procedural and distributional justice issues can play a role in creating discontent about the potential impacts of CCUS projects [e.g. 77, 114, 189]. How impacts were distributed and perceived was related to the social capital of the communities together with their history of marginalisation and empowerment when similar projects had been proposed in their communities [56]. Furthermore, if communities consid-ered that the procedures leading up to the project had been unjust, it could also influence how they perceived the potential risks and benefits of the projects [27]. Building on these arguments was the perspective that the power imbalance between local communities, companies, and the national government should be taken into account when analysing how and why conflicts emerged and how they impacted the outcome of CCUS projects [190]. How to address these issues then become a question of how best to promote procedural, distributional and epistemic justice rather than how to design best practice engagement strategies to communicate the par-ticularities of the project.

To conclude, how impacts are perceived is often an area of conflict between the communi-ties and the particular CCUS project. Although practices of community engagement and com-munication have been found to contribute to shaping understandings of impacts, to what extent they can be divorced from wider social justice issues remains contested.

## Discussion

Our findings illustrate the three main areas of contestation that emerged in the literature on site-specific CCUS projects and social acceptance. We drew out nine dimensions within these areas to explore different approaches within extant work and identify areas for further investi-gation. In this section, we seek to synthesise the commonalities between these approaches to enhance our understanding of community acceptance and social impacts of CCUS projects and consider how these findings might inform future CCUS project practices.

The review found that although there were different and conflicting conceptualisations of community acceptance and impacts, there were still some areas with common ground. These areas of some agreement often tended to cluster around recommendations for community engagement practices for current and future CCUS projects. We identified three recommen-dations where there was partial consensus in the literature: 1) providing transparency; 2) acknowledging uncertainty, and 3) encouraging collaboration.

It is important to note that the rationale for these recommendations might be different depending on how community acceptance and impacts are approached. For example, if accep-tance is conceptualised as a lack of resistance it might still be useful to provide transparency as that would reduce the chance of resistance. Similarly, if acceptance is understood as part of a wider social justice process, transparency can be an important element in these processes. Fur-thermore, although at first glance transparency, uncertainty, and collaboration might be per-ceived as independent recommendations, there are numerous overlaps amongst them and often one leads to another. For example, lack of transparency may lead to more information uncertainty, and lack of collaboration might lead to lack of transparency. Vice versa, collabora-tion might lead to decreased levels of uncertainty and greater levels of transparency.

We also recognise that how exactly these recommendations can and should be interpreted and implemented will depend on the particular context, the project partners, and communities that are connected to particular CCUS projects. Some projects involve activities across several

countries linking partners and communities with different cultural, social, and political characteristics that each influence what is possible in terms of community engagement. Nonetheless, these suggested practices can hopefully help inform beneficial community engagement.

## Transparency

Transparency was considered to be important as it signified open decision making, reduced concerns about corruption and made people and organizations accountable for their actions [191]. Several reviewed studies reported that participants were concerned that either the government or the project owners were not being truthful with different elements of the project. Vattenfall's (Swedish energy company) site in Brandenburg, Germany, has been used in literature as an example of a lack of transparency. Both politicians and the public requested safety-related information from the company, but Vattenfall did not provide them with such information [111]. Individuals were concerned that the company would not publish the results from their exploration work [73], thus reducing their mode of transparency. Another example of a lack of public transparency was Shell's project in Barendrecht, Netherlands, where both the government and the company were perceived as not being transparent about the costs, and impacts associated with the project [29, 75]. Both Brandenburg and Barendrecht projects were cancelled due to public opposition.

Whether a project is deemed to be transparent relates to how trustworthy the project partners are perceived to be and there is no single set of procedures that can guarantee certain outcomes. Nonetheless, the literature suggests that throughout CCUS projects, as much information as possible about the different aspects of the project should be made visible to the public in an easily accessible manner [122]. Importantly, this information should represent a diverse range of views on the piloted technology and the significance of the project [73]. Diverse platforms that are accessible to a variety of groups within communities should be available to the public, e.g., virtual forums, local meetings and consultation events with independent facilitation. Through those platforms, contested questions and concerns can be raised and discussed in the open. These are examples of transparent practices to increase community awareness and acceptance of CCUS projects.

## Uncertainty

Any development of new technologies will involve some element of uncertainty. In other CCUS projects both expert and public uncertainty seems to have been associated with a lack of knowledge and understanding of the technical aspects of CCUS, as well as the functions and processes of ecosystems and the natural environment. From an expert perspective, the uncertainty could be associated with the low technological readiness level (TRL) of CCUS, and the slow pace of development in contrast to that originally planned a decade ago [192]. On the other hand, based on this analysis of pertinent literature, the public appears to be less concerned about how the technology works and more concerned about the unknown processes linking CCUS' potentially negative impacts on the natural environment and public health [111, 168].

How this uncertainty is dealt with partly depends on how project processes are implemented. A project process that plays down uncertainties while promoting expert knowledge as the only useful way of knowing the world risks alienating impacted communities and stakeholders. Similar results may occur from project practices that ignore aspects of climate treaties and, most importantly, miss a learning opportunity to develop and improve the outcomes of the technology. We suggest that recognising the uncertainties surrounding the technology and

opening opportunities to learn from different stakeholders and communities will optimise the management of the inherent uncertainties of innovative CCUS projects.

## Collaboration

Our third and final suggestion is to design and engage in collaboration with communities. Despite the different perspectives on what the purpose of this collaboration should be, whether that is to promote the acceptance of a CCUS project [29], create social learning opportunities [57] or ensure social justice [114], there is some agreement that early and extensive engagement and collaboration are important. For example, if the purpose is to promote acceptance of CCUS, research indicates that engaging comprehensively with communities increases the likelihood of project success compared to projects that do not engage the public [73, 116]. Similarly, if the purpose is to promote social learning expanding collaboration and engagement activities might achieve this [55, 75, 124]. Indeed, some studies demonstrated how a lack of collaboration and comprehensive engagement was seen to exacerbate social patterns of injustice and exclusion [56, 77, 107, 189]. Collaboration can thus not only play an important part in developing more comprehensive levels of community engagement, but it can also contribute to the avoidance of harm within communities impacted by the project. No matter whether these collaborative practices aim to encourage acceptance, social learning, or justice, it is essential to find ways of encouraging and enabling community involvement and collaboration.

## Conclusion

This meta-narrative review has examined research findings on community acceptance and the social impacts of Carbon Capture Utilisation and Storage (CCUS) projects. 53 research papers were identified and analysed according to the meta-narrative principles of pragmatism, pluralism, reflexibility, contestation, historicity, and peer-review.

We found that acceptance, community, and impact were key areas of contestation that had been conceptualised and approached in a variety of ways within the literature. Our analysis identified a further nine dimensions that illustrated the underlying dynamics shaping understandings of acceptance, community, and impacts.

Although the literature highlighted some important aspects of community acceptance, impact and CCUS projects, we identified areas for further investigation. As shown in the results section the majority of CCUS projects, in accordance with this study's inclusion/exclusion criteria that did not consider geographical location, are found in North America, Australia and Europe. India, China and Russia are three countries amongst many that no studies were identified. That is very important to consider as these three countries are amongst the highest $CO_2$ emitters in the world [9]. Although there have been studies on the public's opinions and awareness in China and Russia [193–195], these are not linked to specific CCUS pilot projects. In the case of India, we did not identify any studies on CCUS and social acceptance. Considering the cultural, historical, political, and social complexities that can shape understanding of CCUS, it is important not to generalise findings from some regions as being applicable globally.

Furthermore, we found a lack of attention to the specific characteristics of the technologies used in the projects. As discussed in the results section CCUS encompasses a variety of technologies with different technology levels. These technological differences could also have wider implications for conceptualisations and practices related to social impacts and community acceptance. In future research on CCUS projects, it would therefore be advisable to consider these factors to a greater degree than is evident in the current state of the research body.

Findings also suggest this emerging field of research would benefit from engagement with the wider research on acceptance, community, and impacts, as well as place-based climate change mitigation and adaption initiatives.

Despite these limitations, there are still important lessons to be learned from this review. The relationship between community acceptance, impacts, and CCUS projects is complex, involving many different factors and practices in combinations that may be unique to each project. This means that it is not possible to prescribe precise solutions that will be universally appropriate for all CCUS projects nor is it possible to provide best practice guidelines to ensure particular outcomes. There was however some consensus around some recommended practices that could inform project and community engagement in current and future CCUS projects. These include the provision of transparency regarding the project, together with an acknowledgement of the uncertainties and encouraging collaboration with local communities.

How these shared recommendations from the literature are made sense of and translated into specific project practices will depend on the particular project and we advocate a place-based approach. As demonstrated in the review there is a myriad of contested understandings of community acceptance and impacts at play in CCUS projects and they all tie in with wider social, political, and economical complexities. Nonetheless, our aspiration is that by synthesising the areas of contestation as well as the shared recommendation from the literature this review will help advise future project stakeholders and communities as to how to navigate community acceptance and social impacts in multi-faceted, challenging, and innovative projects.

## Supporting information

**S1 Checklist.**
(DOCX)

**S1 Fig. Bibliometric analysis of literature related to CCUS in general.**
(TIF)

**S2 Fig.**
(DOCX)

**S1 File. Meta-narrative principles.**
(DOCX)

**S2 File. Search process.**
(DOCX)

**S3 File. Search terms.**
(DOCX)

**S4 File. Studies included in the systematic review.**
(DOCX)

## Acknowledgments

We would like to thank our colleagues for providing us with feedback and suggestions to strengthen this manuscript.

## Author Contributions

**Conceptualization:** Jacob A. E. Nielsen, Kostas Stavrianakis, Zoe Morrison.

**Data curation:** Jacob A. E. Nielsen, Kostas Stavrianakis.

**Formal analysis:** Jacob A. E. Nielsen, Kostas Stavrianakis.

**Funding acquisition:** Zoe Morrison.

**Methodology:** Jacob A. E. Nielsen, Kostas Stavrianakis, Zoe Morrison.

**Visualization:** Kostas Stavrianakis.

**Writing – original draft:** Jacob A. E. Nielsen, Kostas Stavrianakis.

**Writing – review & editing:** Jacob A. E. Nielsen, Kostas Stavrianakis, Zoe Morrison.

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
