## [Decision Letter · Decision Letter 0]

9 Mar 2022

PONE-D-21-40197Community acceptance and social impacts of Carbon Capture and Utilization Projects: A systematic meta-narrative literature reviewPLOS ONE

Dear Dr. Nielsen,

Thank you for submitting your manuscript to PLOS ONE. After careful consideration, we feel that it has merit but does not fully meet PLOS ONE’s publication criteria as it currently stands. Therefore, we invite you to submit a revised version of the manuscript that addresses the points raised during the review process. Specifically, we call your attention to the diversity of the CCUS technologies and likewise the public perception, if at all the need be in certain cases as pointed out by Reviewer 1 & 3. In particular, you may categorize the CCUS technologies with respect to the reviews referenced, and present social impacts in view of those categories. While you categorize the CCUS technologies, please make sure to define the categories to provide clarity to the readership and make amendments throughout the manuscript in this respect, as the various terminologies of CCUS employed in this version, confuses the readers. The quality of the figures also needs your special attention. Overall, the manuscript needs a thorough revision before it can be accepted for publication. Please submit your revised manuscript by Apr 23 2022 11:59PM. If you will need more time than this to complete your revisions, please reply to this message or contact the journal office at plosone@plos.org. Please include the following items when submitting your revised manuscript:A rebuttal letter that responds to each point raised by the academic editor and reviewer(s). You should upload this letter as a separate file labeled 'Response to Reviewers'.A marked-up copy of your manuscript that highlights changes made to the original version. You should upload this as a separate file labeled 'Revised Manuscript with Track Changes'.An unmarked version of your revised paper without tracked changes. You should upload this as a separate file labeled 'Manuscript'.

We look forward to receiving your revised manuscript.

Best regards,

Rishiram Ramanan

Academic Editor

PLOS ONE

Journal Requirements:

Reviewers' comments:

Reviewer's Responses to Questions

**Comments to the Author**

1. Is the manuscript technically sound, and do the data support the conclusions?

Reviewer #1: Partly

Reviewer #2: Yes

Reviewer #3: Partly

2. Has the statistical analysis been performed appropriately and rigorously? 

Reviewer #1: N/A

Reviewer #2: Yes

Reviewer #3: Yes

3. Have the authors made all data underlying the findings in their manuscript fully available?

Reviewer #1: Yes

Reviewer #2: Yes

Reviewer #3: Yes

4. Is the manuscript presented in an intelligible fashion and written in standard English?

Reviewer #1: Yes

Reviewer #2: Yes

Reviewer #3: Yes

5. Review Comments to the Author

Reviewer #1: This manuscript presents a summary of literature survey with a focus on the social impact of CCUS projects. The authors identified three major contestations with respect to a broader social discussion around CCUS: acceptance, communities, impacts. Based on these, the authors evaluated the current status of CCUS projects and suggested the importance of transparent and highly collaborative CCUS initiatives, while such initiatives need to acknowledge inevitable uncertainties. Although the manuscript presents some useful discussions, it appears that many of them are too general and therefore one may reach similar suggestions and narratives without extensive literature survey. Specific comments for authors are appended below:

1. Meta-narrative review performed here seemed to be focused on just a handful of studies and it looks like providing more in-depth discussion on the surveyed studies as well as including more relevant studies in the review process will be needed.

2. CCUS technologies are very diverse and each technology is likely to have different social impacts – discussing different social impacts of each CCUS technology will be important in making any public decision around research investments as well as any policies encouraging the spread of effective CCUS technologies.

3. Societal perception towards CCUS technologies will also be hinged on GHG source(s) for which a given technology aims to address (e.g., non-point vs point emissions). For instance, direct air capture (DAC) would have different public acceptance compared to technologies aim to capture carbon emissions from industrial flue gases. Discussions relevant to this point should be added somewhere in the manuscript.

4. How does TRL influence public acceptance towards the corresponding technology? Even though CCUS generally has a low TRL, it will be interesting to know if there is a certain TRL threshold at which public engagement becomes more important. On another note, CCUS technologies will require a long-term R&D investment – it is therefore possible that present public acceptance of a specific technology may not lead to its successful future implementation. How one can maintain a broad project portfolio for long-term CCUS initiatives while successfully engaging public sectors?

Reviewer #2: This article provides an overview of publications related to community acceptance of CCUS project implementation by using a meta-narrative approach. The article is informative and logical, and the arguments in the manuscript are instructive for scholars in the field of CCUS acceptance. I think the following questions can be revised to further improve the quality of the manuscript and meet the requirements for publication in this journal.

1. In lines, 66-65, the data for CO2 emissions are from 2017 and it is recommended to use the latest year of CO2 emissions data.

2. In line 75, A full stop is missing after (15).

3. The font and clarity of Figure 1 could be suitably adjusted to make it more aesthetically pleasing as well as more readable.

4. The layout of Fig. 4 is not well designed and there is too much blank space in the right half, please make reasonable adjustments to make it more aesthetically pleasing.

5. Adjusting the size of the box and font in Fig. 6.

6. Please check whether it is "Figure 6" or "Figure 6." to make the name of the figure consistent with the previous text.

Reviewer #3: Dear Authors,

Thank you for an very interesting manuscript.

(1) My main questions/concern is which project are you exactly studying/assesssing. In the title you state CCU and in the main body you refer to CCUS, with emphasis on CCS. However, for which stage of a CCUS project do you need the public acceptance? If an existing company decided to install a carbon capture facility in one of their plants (as a private investement), why does this need to be a matter of public acceptance. It is a private decision from the part of the company. If a company decides to use recycled CO2 in order to produce e.g. methanol, instead of fossil CO2, why does the public needs to give their acceptance? Again, it is a private decision. The public might decide not to buy the new methanol, because of certain reasons, but should they have an opinion about private company investments.

The public should be involved when new transportation infrastructure will be built or when CO2 will be stored somewhere near their community. So, are you reviewing public acceptance of CCS? Carbon transporation? CCU products? In my opinion, the approach should be different between these cases. For example, in the case of products, no consultation makes sense. When it comes to the market, the public may buy it or not.

You should define better what exactly you are assessing.

(2) Another question is what do you mean by CCUS technologies? Which technologies are you assessing? Capture Technologies? Alternative uses? Transporation Methods? Are there any specific storage technologies of importance?

(3) Your reference list is mixed up. I tried finding a few references but couldn't.

(4) See also comments in the attached.

Yours Sincerely

6. PLOS authors have the option to publish the peer review history of their article (what does this mean?). If published, this will include your full peer review and any attached files.

Reviewer #1: **Yes: **Jin-Ho Yun

Reviewer #2: No

Reviewer #3: No

---

## [Author Response · Author response to Decision Letter 0]

4 May 2022

Response to reviewers (see also attached response to reviewers letter).

Community acceptance and social impacts of Carbon Capture, Utilization and Storage Projects: A systematic meta-narrative literature review

Dear reviewers and Rishiram Ramanan,

We would like to express our gratitude to all of the reviewers and editor for providing feedback on our article and for the many helpful suggestions on how to improve its quality. We have made the following adjustments to the manuscript to improve our paper based on this feedback. 

Reviewer 1

1. Meta-narrative review performed here seemed to be focused on just a handful of studies and it looks like providing more in-depth discussion on the surveyed studies as well as including more relevant studies in the review process will be needed.

Response: We recognise that whilst a meta-narrative systematic review can give an overview of research contestations and highlight overall gaps in the research field, it is not adapt at exploring the fine-grained details of the research papers. We emphasised this point further at P6 (L123). We believe that focusing on research related to site-specific CCUS projects provides insights that can not be learned by expanding the definition to include non-specific research as is also discussed in P5 (L102).

2. CCUS technologies are very diverse and each technology is likely to have different social impacts – discussing different social impacts of each CCUS technology will be important in making any public decision around research investments as well as any policies encouraging the spread of effective CCUS technologies. 

Response: We agree that the specifics of the CCUS technologies and their technological readiness level can have consequences on how practices and conceptualisations of community acceptance and social impact are shaped. However, these aspects were largely left unexplored in the papers that were reviewed both in terms of specifying exactly what technology was applied and in terms of how this relates to social impacts and community acceptance. Given the current state of the literature on this aspect it is not possible to categorise CCUS categories in relation to their social impact and community acceptance. In the results section P14 (L282) and the conclusion P47 (L1105), we have added some paragraphs to touch upon these limitations. We have also addressed updated figure 4 on page 14 on the different stages of CCUS as well as the summary list of the used studies in supplementary material S5 on page 54. Finally, on P8 (L172), we have added some context to our discussion on why we have used CCUS as a catch-all term to encompass the different technologies associated with carbon capture, utilisation, and storage. 

3. Societal perception towards CCUS technologies will also be hinged on GHG source(s) for which a given technology aims to address (e.g., non-point vs point emissions). For instance, direct air capture (DAC) would have different public acceptance compared to technologies aim to capture carbon emissions from industrial flue gases. Discussions relevant to this point should be added somewhere in the manuscript.

Response: This is a very valid point. The reviewed literature did not include any studies onsite specific DAC projects which we have specified at P14 (L289). As mentioned in the response above we have added some paragraphs to address the issue of the diversity of CCUS technologies. 

4. How does TRL influence public acceptance towards the corresponding technology? Even though CCUS generally has a low TRL, it will be interesting to know if there is a certain TRL threshold at which public engagement becomes more important. On another note, CCUS technologies will require a long-term R&D investment – it is therefore possible that present public acceptance of a specific technology may not lead to its successful future implementation. How one can maintain a broad project portfolio for long-term CCUS initiatives while successfully engaging public sectors?

Response: These are all interesting points, but unfortunately the maturity of the literature does not make it possible to make such conclusions regarding TRL at the moment. We have however added some paragraphs to discuss these limitations as is also mentioned above. In terms of the importance of taking account of market factors, we have added some discussion on market acceptance on P20 (L404) and P22 (L452).

Reviewer 2

1. In lines, 66-65, the data for CO2 emissions are from 2017 and it is recommended to use the latest year of CO2 emissions data.

Response: In lines 65-66 we have provided data from 2021 instead of 2017

2. In line 75, A full stop is missing after (15).

Response: We added a full stop (after revision now it's L74).

3. The font and clarity of Figure 1 could be suitably adjusted to make it more aesthetically pleasing as well as more readable.

Response: We have adjusted the font and clarity of Figure 1. 

4. The layout of Fig. 4 is not well designed and there is too much blank space in the right half, please make reasonable adjustments to make it more aesthetically pleasing.

Response: We changed the layout so there is less blank space in the right half of the figure.

5. Adjusting the size of the box and font in Fig. 6.

Response: The font has been increased to make it more readable.

6. Please check whether it is "Figure 6" or "Figure 6." to make the name of the figure consistent with the previous text.

Response: We have adjusted the figures do not include a full stop after the number.

Reviewer 3

1. My main questions/concern is which project are you exactly studying/assesssing. In the title you state CCU and in the main body you refer to CCUS, with emphasis on CCS. However, for which stage of a CCUS project do you need the public acceptance? If an existing company decided to install a carbon capture facility in one of their plants (as a private investement), why does this need to be a matter of public acceptance. It is a private decision from the part of the company. If a company decides to use recycled CO2 in order to produce e.g. methanol, instead of fossil CO2, why does the public needs to give their acceptance? Again, it is a private decision. The public might decide not to buy the new methanol, because of certain reasons, but should they have an opinion about private company investments.

The public should be involved when new transportation infrastructure will be built or when CO2 will be stored somewhere near their community. So, are you reviewing public acceptance of CCS? Carbon transporation? CCU products? In my opinion, the approach should be different between these cases. For example, in the case of products, no consultation makes sense. When it comes to the market, the public may buy it or not.

You should define better what exactly you are assessing.

Response: There were some grammatical mistakes in the title and throughout the document where we referred to CCS instead of CCUS which has now been corrected. On P8 (L172), we have added some context to our discussion on why we have used CCUS as a catch-all term to encompass the different technologies associated with carbon capture, utilisation, and storage. 

In relation to the issue of whether public acceptance is necessary, we concur that to what extent community acceptance is necessary is debated and can depend on other market and political-related circumstances. To highlight this we have added some paragraphs and details on this aspect in the acceptance section P20 (L404) and P22 (L452). 

2. Another question is what do you mean by CCUS technologies? Which technologies are you assessing? Capture Technologies? Alternative uses? Transporation Methods? Are there any specific storage technologies of importance?

Response: The inclusion criteria were kept broad in that any site-specific paper that related to community acceptance of any part of the CCUS technology was included. The inclusion criteria can be accessed in the methods chapter and S3 and S4. We agree that the complexity and diversity of technologies under the term “CCUS” can have divergent consequences for community acceptance and social impacts. However, these aspects were largely left unexplored in the papers that were reviewed both in terms of specifying exactly what technology was applied and in terms of how this relates to social impacts and community acceptance. Given the current state of the literature on this aspect it is not possible to categorise CCUS categories in relation to their social impact and community acceptance. In the results section P14 (L282) and the conclusion P47 (L1105), we have added some paragraphs to touch upon these limitations. We have also addressed updated figure 4 on page 14 on the different stages of CCUS as well as the summary list of the used studies in supplementary material S5 on page 54.

3. Your reference list is mixed up. I tried finding a few references but couldn't.

Response: We have updated the references throughout as there was an error in the management software. These changes are not all visible on the track change system in MS Word as there were some compatibility issues between the citation management software and the use of track changes (if the track changes had been on, all paragraphs with a citation would seem like it had been changed completely).

4. See also comments in the attached.

Response to attached comments:

Reviewer: The title does not agree with the text. In the text you mention CCUS, whereas in the title you have only CCU.

Response: Changed to "Community acceptance and social impacts of Carbon Capture, Utilization and Storage Projects: A systematic meta-narrative literature review.

Reviewer: The repetition here could be avoided.

Response: Changed to "design and implementation". Line 14

Reviewer: I would advise avoiding 1st person.

Response: Addressed it throughout the document.

Reviewer: What do you mean? Original document line 19

Response: Added “site specific CCUS studies”. Line 19

Reviewer: Use a more appropriate verb. Original document line 20

Response: Rewrote the paragraph. Line 19-29

Reviewer: Which stage of the CCUS project are you assessing? Capture, Utilization or Storage? Each stage has different impact in the local community and different regional characteristics. You cannot assess them with the same analysis.

Response: Please see response to comment one and two.

Reviewer: Is that the appropriate noun? Original document line 33

Response: Rewrote sentence starting Line 26.

Reviewer: degrees Celsius

Response: Changed to “degrees Celsius” Line 49

Reviewer: It is in 2019 so per year not needed

Response: “Per year” removed. Line 72.

Reviewer: How do you define larger projects? 

Response: Added” greater than 0.3 Mt CO2 per year”. Line 76

Reviewer: What did this add? Did you apply it to the 14 extra studies or something else?

Response: Added explanation in Figure 1. Line 204.

Reviewer: In Figure 1 and Figure 2 you mention 1997. Explain the difference? No papers before 2009 were site specific?

Response: There were no site specific papers prior to 2009. Figure 2 has been moved to S2 and is referred to in S1 in relation to the operationalisation of reflexivity. 

Reviewer: Not all stages of CCUS are deployed in the same community. Which community are you talking about? The region near the capture plant or near the storage?

Response: Added, “Proposed CO2 storage project”. Line 360.

Reviewer: (61) is Wong G, Greenhalgh T, Westhorp G, Buckingham J, Pawson R. RAMESES publication standards: meta-narrative reviews. 2013;15. This is not about CCUS in Australia. Please double check references.

Response: Addressed the citations throughout the document.

Reviewer: Similarly (64) is Vercelli S, Lombardi S. CCS as part of a global cultural development for environmentally sustainable energy production. Gale J, Herzog H, Braitsch J, editors. Greenh GAS CONTROL Technol 9. 2009.This is not about California. Check you reference list.

Response: Addressed the citations throughout the document.

Reviewer: Zerogen is (72)

Response: Addressed the citations throughout the document.

Reviewer: a. Original document line 378

Response: Changed to “A”. Line 417.

Reviewer: What do you mean by new energy?

Response: Rewrote sentence Line 515.

Reviewer: What are the CCUS technologies?

Response: Rewrote sentence line 563.

Reviewer: What do you mean by other CCUS processes? Does the community have to accept these other processes/stages of CCUS?

Response: Added"CO2 whether that was in projects that focused exclusively on the storage stage or whether it was in connection with studies that also explored the carbon capture, transportation, and/or utilisation stage of CCUS. (See figure 4 in the results section)." Line 700.

Reviewer: Rarely does a project includes both U and S. It is either utlization or storage. Please make that clear.

Response: added "Although there are no comprehensive databases that encompass pilot and demonstration sites (16) commercially operating CCUS facilities have so far been limited to capture and storage (152). However, many of the ongoing and upcoming demonstration and pilot projects involve the utilisation element and one database estimates that 139 ongoing or planned projects involve a utilisation stage whereas the same number for projects that focuses solely on the capture and storage part is estimated to be 192 (153). Although capture and storage continue to be prevalent in CCUS projects there are also indications that the utilisation element could become more significant. It is therefore important to pay attention to how these evolving technologies might also change how relevant communities are conceptualised and located as well as how engagement practices are shaped.

" Line731.

Reviewer: What about storage?

Response: Added “storage”. Line 801.

Reviewer: Ref style. Original document line 780

Response: Addressed throughout the document.

Reviewer: “?” Original text line number 923

Response: Added, “ be an important element in these processes”. Line 995.

---

## [Decision Letter · Decision Letter 1]

14 Jun 2022

PONE-D-21-40197R1Community acceptance and social impacts of Carbon Capture, Utilization and Storage Projects: A systematic meta-narrative literature reviewPLOS ONE

Dear Dr. Nielsen,

Thank you for submitting your manuscript to PLOS ONE. After careful consideration, although reviewers have consented to accept the manuscript for publication, we feel that the manuscript does not fully meet PLOS ONE’s publication criteria as it currently stands. Therefore, we invite you to submit a revised version of the manuscript that addresses the points raised by the Academic Editor.

We look forward to receiving your revised manuscript.

Kind regards,

Rishiram Ramanan

Academic Editor

PLOS ONE

Journal Requirements:

Additional Editor Comments:

Dear Authors,

Although the reviewers have consented to accept the article for publication, there are minor revisions that need to be performed to maintain the publication standards set by PLOS One. My concern is the quality and consistency of the figures.

1. Some figures use major and minor grid lines, while some figures use major grid lines and others have none. Is there any specific reason for the same, as all these figures deal with number of publications/ studies in the y-axis?

2. Similarly, the title within the figure, apart from the figure caption is unnecessary.

3. Certain bar graphs have shadows, and gradient backgrounds, which are not consistent with other figures.

4. Do not include figures in the main manuscript file. Each figure must be prepared and submitted as an individual file. Please refer to the submission guidelines in the PLOS One website, and please improve the overall quality of the said figures.

Academic Editor

Reviewers' comments:

Reviewer's Responses to Questions

**Comments to the Author**

1. If the authors have adequately addressed your comments raised in a previous round of review and you feel that this manuscript is now acceptable for publication, you may indicate that here to bypass the “Comments to the Author” section, enter your conflict of interest statement in the “Confidential to Editor” section, and submit your "Accept" recommendation.

Reviewer #1: All comments have been addressed

Reviewer #2: (No Response)

Reviewer #3: All comments have been addressed

2. Is the manuscript technically sound, and do the data support the conclusions?

Reviewer #1: Yes

Reviewer #2: (No Response)

Reviewer #3: Yes

3. Has the statistical analysis been performed appropriately and rigorously? 

Reviewer #1: N/A

Reviewer #2: (No Response)

Reviewer #3: Yes

4. Have the authors made all data underlying the findings in their manuscript fully available?

Reviewer #1: Yes

Reviewer #2: (No Response)

Reviewer #3: Yes

5. Is the manuscript presented in an intelligible fashion and written in standard English?

Reviewer #1: Yes

Reviewer #2: (No Response)

Reviewer #3: Yes

6. Review Comments to the Author

Reviewer #1: (No Response)

Reviewer #2: (No Response)

Reviewer #3: (No Response)

7. PLOS authors have the option to publish the peer review history of their article (what does this mean?). If published, this will include your full peer review and any attached files.

Reviewer #1: No

Reviewer #2: No

Reviewer #3: No

---

## [Author Response · Author response to Decision Letter 1]

13 Jul 2022

Dear reviewers and academic editor

Thank you to the reviewers for their previous feedback and for accepting our revised manuscript.

Thank you also to the academic editor for your feedback on bringing the manuscript up to the standards for Plos One. We have made the following adjustments to respond to your feedback.

Academic Editor

1. Some figures use major and minor grid lines, while some figures use major grid lines and others have none. Is there any specific reason for the same, as all these figures deal with number of publications/ studies in the y-axis?

Response: We have standardized all the figures so they use the same kind of format.

2. Similarly, the title within the figure, apart from the figure caption is unnecessary.

Response: We have removed the title within the figures.

3. Certain bar graphs have shadows, and gradient backgrounds, which are not consistent with other figures.

Response: We have standardized the figures so they have the same format

4. Do not include figures in the main manuscript file. Each figure must be prepared and submitted as an individual file. Please refer to the submission guidelines on the PLOS One website, and please improve the overall quality of the said figures.

Response: We have followed the guidelines including removing the figures from the manuscript file and uploading the files separately as individual files. We have also uploaded the files to Pace to ensure quality compliance. We have also changed the headings to follow guidelines (e.g Heading Level 1 18pt, Level 2 16pt, Level 3 14 pt.)

Kind regards,

The authors

---

## [Editor Report · Decision Letter 2]

20 Jul 2022

Community acceptance and social impacts of Carbon Capture, Utilization and Storage Projects: A systematic meta-narrative literature review

PONE-D-21-40197R2

Dear Dr. Nielsen,

We’re pleased to inform you that your manuscript has been judged scientifically suitable for publication and will be formally accepted for publication once it meets all outstanding technical requirements.

Best regards,

Rishiram Ramanan

Academic Editor

PLOS ONE
---

## [Editor Report · Acceptance letter]

25 Jul 2022

PONE-D-21-40197R2 

Community acceptance and social impacts of Carbon Capture, Utilization and Storage Projects: A systematic meta-narrative literature review 

Dear Dr. Nielsen:

I'm pleased to inform you that your manuscript has been deemed suitable for publication in PLOS ONE. Congratulations! Your manuscript is now with our production department. 

Kind regards, 

on behalf of

Dr. Rishiram Ramanan 

Academic Editor

PLOS ONE